# Post-COVID-19 Status and Its Physical, Nutritional, Psychological, and Social Effects in Working-Age Adults—A Prospective Questionnaire Study

**DOI:** 10.3390/jcm11226668

**Published:** 2022-11-10

**Authors:** Tamara Nikolic Turnic, Ivana Vasiljevic, Magdalena Stanic, Biljana Jakovljevic, Maria Mikerova, Natalia Ekkert, Vladimir Reshetnikov, Vladimir Jakovljevic

**Affiliations:** 1Department of Pharmacy, Faculty of Medical Sciences, University of Kragujevac, 34000 Kragujevac, Serbia; 2N.A. Semashko Public Health and Healthcare Department, F.F. Erisman Institute of Public Health, I.M. Sechenov First Moscow State Medical University, 119435 Moscow, Russia; 3Academy for Applied Studies, The College of Health Studies, 11070 Belgrade, Serbia; 4Department of Physiology, Faculty of Medical Sciences, University of Kragujevac, 34000 Kragujevac, Serbia; 5Department of Human Pathology, 1st Moscow State Medical, University IM Sechenov, Trubetskaya Street 8, Str. 2, 119991 Moscow, Russia

**Keywords:** post-COVID-19 status, physical activities, nutritional habits, social activity, working-age adults

## Abstract

Background: The main objective of this study was to evaluate the evolution of physical and daily routine, dietary habits, and mental and social health in individuals with recent COVID-19 infection. Methods: A qualitative prospective cross-sectional study was conducted from 01 October 2021 to 01 March 2022, which included 80 working-age adults from the territory of Central Serbia who had PCR-confirmed SARS-CoV-2 infection in the previous six months. Two structured pre-coded closed-ended questionnaires were submitted to the participants: a questionnaire about post-COVID-19 status (pCOVq) and a shortened version of the World Health Organization’s Quality of Life Scale (WHOQOL-BREF). Results: The presence of the COVID-19 disease in the previous period of 6 months among the working-age participants significantly affected the duration of aerobic, anaerobic, and high-intensity physical activities, but also the possibility of performing certain activities such as walking, which represents basic aerobic activity and a measure of general health among middle-aged participants. In the majority of cases (78%), in the post-COVID-19 period, participants indicated a decline in educational and productive activities. Conclusion: Post-COVID status in working-aged participants consists of reduced physical activity, lower quality of life, and similar nutritional habits. Health policies should be more focused on these findings.

## 1. Introduction

After the December outbreak of COVID-19 in Wuhan, China, the infection spread rapidly to other parts of the world, and the increasing number of cases suggests that the disease continues to spread steadily [1]. Initially, it was thought that people who had been to the seafood market or consumed food prepared with infected animals had been infected with COVID-19. However, it was later found that a number of people who had not traveled to the seafood market were also infected, i.e., tested positive for COVID-19 [2,3]. This showed that human-to-human transmission of the virus was possible, which was later confirmed in many countries around the world [1,2,3,4]. Thanks to the interventions and control measures taken by governments around the world, the number of newly confirmed cases is decreasing worldwide [5]. Although all important measures are being implemented, the risk of transmission has not yet been eliminated, and the epidemic remains a major challenge for clinicians. A large proportion of patients have a mild or moderate course of disease, but a number of patients have a severe and even life-threatening course of disease [6,7].

Social distancing and confinement have been very important in the fight against the spread of the virus [8]. However, all these restrictions affect the health and well-being of the general population. In these circumstances, some sudden and stressful situation after a long stay at home can greatly affect the change of lifestyle, i.e.: physical activity, eating habits, alcohol consumption, mental health, quality of sleep, etc. [9,10].

Further, recommended travel restrictions and regulations against participation in outdoor activities, which includes physical activity and exercise, have greatly affected the routine daily activities of a wide variety of people [11].

Post-COVID syndrome was described as a clinical entity for the first time in the spring of 2020 when patients with COVID-19 still had symptoms several weeks after acute infection [12]. Currently, this syndrome is one of the challenges for health systems, which is becoming more common as the pandemic develops. Recent studies suggest that 10 to 20% of patients with SARS-CoV-2 who go through the acute symptomatic phase have disease sequelae in the form of unexplained, persistent signs or symptoms for 12 weeks after SARS-CoV-2 infection, called “post-COVID syndrome”. In general, patients with this syndrome had experience fatigue, extreme tiredness, and symptoms that persist beyond the active phase of the disease [13,14,15].

Although many patients recover without sequelae, many patients may have symptoms for a long time after recovering from the infection, and others may develop new symptoms. The most common persistent symptoms in the post-infection period (more than 4 weeks from onset) are shortness of breath, cognitive impairment, fatigue, anxiety, and depression [13,14,15].

The cause of this post-viral syndrome remains unknown, although it resembles chronic fatigue syndrome, which is now referred to as post-viral fatigue syndrome. If these symptoms are simply due to critical illness or hypoxia in ventilator-dependent patients, this would not explain the reason why they occur in non-hospitalized patients and why they are not clearly related to the severity of the original infection [14].

Future research is very important to explain the pathogenesis, clinical spectrum, and prognosis of post-COVID syndrome. In addition, markers that allow for the rapid diagnosis of this syndrome and monitoring of its associated morbidity and prognosis are needed.

The main objective of this study was to assess the evolution of physical and daily routine, as well as mental and social health in individuals with a recent COVID-19 infection over a six-month period after infection and to identify factors associated with (unfavorable) evolution. Secondary objectives were to assess general health and vital energy in working-age individuals, social functioning, and dietary behaviors in the period after COVID.

## 2. Materials and Methods

### 2.1. Ethics Approval and Consent to Participate

This study protocol was approved by the Institutional Ethics Committee of the Faculty of Medical Sciences of the University of Kragujevac (No. 147/20) and is in accordance with the principles of the Declaration of Helsinki (2013 revision). Participation in this study was voluntary and anonymous.

### 2.2. Protocol of the Study

A qualitative prospective cross-sectional study was conducted from 1 October 2021 to 1 March 2022, involving 80 working-age adults from the area of central Serbia who had PCR-confirmed SARS-CoV-2 infection in the previous six months.

### 2.3. Recruitment of Participants

The inclusion criteria were the presence of SARS-CoV-2 infection in the previous 6 months from inclusion in the study (positive PCR test), ambulatory treatment during COVID-19 disease, an absence of other chronical diseases/treatments, place of residence in the cities of central Serbia, and a positive answer to the question from the first part of the questionnaire (voluntary consent to the study). Exclusion criteria were age less than 18 years and more than 50 years old, place of residence in other countries/regions of the world, as well as an incompletely completed survey and previous hospital treatment regarding COVID-19 disease.

### 2.4. Instruments

Two structured pre-coded closed-ended questionnaires were submitted to the participants: a questionnaire about post-COVID-19 status (pCOVq) and a shortened version of the World Health Organization’s Quality of Life Scale (WHOQOL-BREF).

### 2.5. A Questionnaire about the Post-COVID-19 Status (pCOVq)

The questionnaire on the post-COVID-19 status of adults in the territory of the municipality of Kragujevac is a standardized tool constructed by the author Kang-Hyun Park and associated and validated on the population of adult respondents during 2020 [16]. The pCOVq is an anonymous questionnaire that consists of 26 questions; each question has up to 6 sub-questions related to the period before and after the infection, and the time allotted for filling it is less than 60 min. The questionnaire consisted of domains, each of which separately analyzed one aspect of life related to the patient, i.e., the respondent (1) physical activity, (2) participation in activities, and (3) nutrition. In previous studies, the pCOVq questionnaire showed high internal reliability, with a Cronbach alpha of 0.83. The intraclass correlation coefficient was 0.97 for the total score of the questionnaire in terms of test–retest reliability [16].

### 2.6. Physical Activity as a Lifestyle Indicator in Post-COVID Period

A total of six physical activity items were assessed using a five-point Likert scale to measure respondents’ frequency of participation in six different physical activities and their satisfaction with participating in these physical activities. The six physical activities included aerobic physical activity; anaerobic physical activity; low-intensity physical activity equivalent to 2–2.9 metabolic equivalents of the task (MET), including gardening, house cleaning, etc.; moderate-intensity physical activity equivalent to 3–5.9 METs, including swimming, doubles tennis, etc.; high-intensity physical activity equivalent to 6–9.9 MET, such as running, climbing, etc.; and walking exercises. According to the American College of Sports Medicine (ACSM), physical activity can be divided into three types, based on intensity. To assess the impact of the COVID-19 pandemic on physical activity, questions were asked about the frequency and satisfaction of participating in physical activity weekly before and after the onset of COVID-19. The higher the score, the higher the level of participation and satisfaction with physical activity.

### 2.7. Daily Activity as a Lifestyle Indicator in POSt-COVID Period

Six items about the daily activity were also assessed using a five-point Likert scale to measure the frequency of and satisfaction with the variety of present daily activities, such as activities of daily living (ADL), leisure, social activities, work, education, and sleep during the week before and after COVID-19. A higher score indicates more frequent participation in various activities as well as greater satisfaction with participation in activities.

### 2.8. Nutrition as a Lifestyle Indicator in Post-COVID Period

Finally, nine items on diet were assessed using a five-point Likert scale to assess diet in the week before and after COVID-19 and to measure participants’ nutritional status. The amounts of carbohydrates, proteins, fats, vitamins, minerals, water, and alcohol that participants consumed, as well as the frequency of drinking and smoking, were measured. For example, participants were asked, “Before the COVID-19 pandemic, how often did you consume carbohydrate-rich foods such as rice, bread, and flour in the last week?” Participants responded to these questions by selecting one of the five-point Likert scale options: (1) never, (2) 1–2 times a week, (3) 3–4 times a week, (4) 5–6 times a week, and (5) every day. A higher score indicates a higher consumption of each type of food.

### 2.9. A Shortened Version of the World Health Organization’s Quality of Life Scale (WHOQOL-BREF)

A shortened version of the World Health Organization’s Quality of Life Scale (WHOQOL-BREF) was used to assess quality of life [17,18]. It contains 26 items rated on a five-point Likert scale and measures four domains: physical, psychological, social and environmental. Raw domain scores were converted to a scale ranging from 0 to 100 to facilitate comparison with other instruments, with higher scores indicating higher quality of life. Cronbach’s alpha was 0.91 and 0.94 before and after COVID-19, respectively. Each question has 7 possible answers scored on a Likert scale from 0 to 6, with 0 corresponding to the feeling that it “never” happens and 6 indicating that it happens “every day”. Quality of life was assessed only six months after COVID-19, since we could not predict who will be infected [17,18].

### 2.10. Data Management and Statistical Analysis

This cross-sectional prospective study included the participants who satisfied the inclusion criteria mentioned before. According to the results of the previous study, the sample size was arrived at using the margin of error approach as seen in the equation below: *n* = Z2P(1−P)/d2. The sample size was set at 80 participants. In the evaluation of the quality of life and clinical status, 4 (5%) participants were excluded because of an incompletely filled survey, which is permissible exclusion. Statistical analyses were performed in the SPSS program version 26 using descriptive and statistical analytical tests. Results are presented as frequencies in percent (%) or as arithmetic means and standard deviations (X; SD). A Chi-square test was used to estimate the statistical differences in the distributions of categorical variables. The statistical threshold was set at the level of 0.05.

## 3. Results

### 3.1. Basic Demographic Characteristics of Study Population

Table 1 shows the basic demographic characteristics of the study population. Most participants were female, with a mean age of 30.64 ± 1.54 years. In addition, most participants were under 30 years of age, single, employed, and living in a medium-sized city (Table 1).

### 3.2. Clinical Status of the Working-Age Study Population

In the form of Table 2, the clinical symptoms among the study population during the period after COVID-19 are shown. During COVID-19, most patients had many current symptoms such as sleep problems, bowel and bladder dysfunction, and respiratory problems. Some of the participants had joint movement restrictions, odor disturbances, circulatory problems, and pain. Interestingly, in the period after COVID, many of these complaints were still present, such as limitations in the sense of smell (17.1%) and taste (15.8%) and breathing problems (14.5%), and most participants had persistent increased fatigue (23.7%). Further, there was a not insignificant number of participants with other persistent symptoms six months after COVID-19 (Table 2).

### 3.3. Changes in Different Types and Intensities of Physical Activity in the Working-Age Study Population after SARS-CoV-2 Infection

In this study, we examined the effects of SARS-CoV-2 infection on the different types and intensities of physical activity six months after the onset of the disease (Table 3). We asked participants about the frequency of aerobic, anaerobic, daily, light, and high-intensity physical activity. Interestingly, both aerobic and anaerobic activity changed, and a greater number of participants reduced these physical activities in the period after COVID. Similarly, the number of participants who engaged in high-intensity physical activity decreased, whereas the consumption of low- and moderate-intensity physical activity did not decrease (Table 3).

In terms of duration, there was an increase in the number of participants who did not engage in physical activity during the period after COVID. The opportunity to engage in physical activity after SARS-CoV-2 infection was also reduced, especially for aerobic and high-intensity activities (Table 3).

Regarding the other activities, Table 4 shows the statistical differences in the period before and six months after COVID-19 in the study population. Statistically significant reductions were observed in the frequency and duration of daily and social activities and in the duration of productive and educational activities. As expected, social activities were most reduced in our working-age population (Table 4).

### 3.4. Nutrition Habits in the Post-COVID Period among Study Population

From Figure 1 and Table 5, we can see that dietary habits changed significantly in the period after COVID, although most of the changes were not statistically significant. Interestingly, the frequency of water consumption changed in the period after COVID, with a large number of participants consuming water five to six times per week or daily (Figure 1; Table 5). In general, the National Academies of Sciences, Engineering, and Medicine suggests that each day, women drink a total of about 2.7 L (L) or 11 cups of fluid and men consume about 3.7 L (16 cups). Every additional cup is recognized as additional water consumption (Figure 1; Table 5).

### 3.5. Quality of Life of Working-Age Participants after SARS-CoV-2 Infection

Table 6 shows the frequency of responses related to self-satisfaction and quality of life. Most participants were satisfied in many ways, but there are still a large number of participants who were generally not satisfied. All questions refer to post-infection symptoms (COVID), so we found that a statistically significant number of participants were only dissatisfied to very dissatisfied with life, health status, daily activities, and personal relationships, which can be considered as decreased satisfaction (Table 6).

## 4. Discussion

The primary objective of this study was to assess the evolution of physical and daily routine, as well as mental and social health in individuals with recent COVID-19 infection over a six-month period after infection and the factors associated with (un)favorable evolution. Secondary objectives were to assess general health and vital energy in working-age individuals, social functioning, and dietary behaviors in the post-COVID period.

For the first time, a study addresses the assessment of the impact of acute infection on the range of other vital activities in young people of working age [19,20]. A variety of questionnaires were used to assess behavior, satisfaction, and objective interference with physical activities. The identification of risk factors for prolonged disease progression is necessary, and early identification and monitoring of patients at increased risk for developing post-syndromal COVID must be a priority.

It has been proven that more than 60% of patients who had COVID-19 have many persistent symptoms and syndromes in the period after COVID [21]. This information could be important not only for healthcare professionals, but also for the economy and people in general, because protracted illness is a global problem, not just an individual disability.

Post-COVID syndrome is a serious condition that includes a variety of new, recurrent, or persistent symptoms that people usually experience at least four weeks after a positive SARS-CoV-2 test [22]. In some cases, symptoms persist for more than 3 months when this syndrome begins as a chronic form of the disease [23].

The results of our study show that there is a statistically significant difference in certain segments before and after infection. Regarding aerobic and anaerobic exercise, the number of subjects who performed these activities decreased significantly after infection. A statistically significant difference was found in the performance of low- and high-intensity physical activities before and after infection.

The duration of low-intensity physical activity decreased sharply, and for high-intensity physical activity, the duration, frequency, and opportunity to engage in it decreased after infection (Table 2, Table 3 and Table 4). There was also a difference in the duration and opportunity to walk before and after infection, with a decrease in the length of walking and an increase in the number of respondents who reported walking as much as possible. Responses on activities of daily living and leisure showed a statistically significant difference in the performance of these activities before and after infection. The frequency with which daily activities were performed decreased, as did the duration of leisure activities. There was a difference in the frequency with which social and educational activities were performed and in the opportunity to perform them (Table 4). The questions about productive and social activities showed a difference in the duration of these activities, with a significant decrease in the number of respondents who engaged in sports activities for more than one hour. Regarding dietary habits before and after infection in terms of carbohydrates, proteins, vitamins, minerals, and fatty foods, statistical analysis showed that there was no statistically significant difference in the consumption of these nutrients (Figure 1, Table 4).

Regular physical activity is one of the most effective ways to prevent morbidity and early death. The World Health Organization recommends at least 150 min of moderate physical activity per week [24,25]. Individual factors such as gender, age, and health status influence the physical activity that people engage in. During the COVID-19 pandemic, a large proportion of the population remained confined to their homes [24,25]. To prevent physical inactivity, experts recommended getting up, walking around the house, and exercising online. During the pandemic, there were negative effects on physical activity intensity and an increase in the consumption of less healthy foods, as well as an increase in sedentary lifestyle. A decrease in physical activity was also observed in college students, but an increase in anxiety was also observed in individuals aged 18–34 years [26,27].

A study by Bakhsh et al. [24] was conducted with the aim of determining whether the dietary and physical activity patterns of the adult Saudi population changed during the COVID-19 quarantine. The methodology of the study was based on an electronic questionnaire that assessed changes in body weight, eating habits, and physical activity among the adult population of Saudi Arabia (*n* = 2255) during the COVID-19 quarantine and was disseminated through social networks during the period between June and July 2020. The results of this study showed that over 45% of participants reported consuming larger amounts of food and snacks, which led to weight gain in approximately 28% of subjects. Feeling bored and empty, as well as the availability of time to prepare meals, were the main reasons for changing eating habits. Honey and vitamin C were the most commonly consumed foods to boost immunity. COVID-19 negatively affected physical activity in 52% of respondents, which was associated with a significant increase in body weight (*p* < 0.001) [21]. The aforementioned study examined the changes in dietary habits of the adult population of Saudi Arabia from the aspect of respondents’ weight gain, whereas our study examined the difference in micro- and macronutrient intake before and after Corona virus infection. However, our results did not show that there was a statistically significant difference in the intake of macro- and micronutrients analyzed in our questionnaire.

A study by the author Ferreira Rodrigues et al. [25] was conducted with the aim of showing changes in food consumption and food product choices during the pandemic COVID-19 in the Brazilian population. Consumer perceptions of food safety and marketing issues were also studied. An online survey was conducted, and the data were analyzed using descriptive analysis. The results of the study show that the COVID-19 pandemic has affected the consumption and purchase of food. Respondents reported eating and buying larger quantities of food, indicating an increased tendency to eat less healthy food, especially among women. On the other hand, homemade preparations and fresh foods were preferred [25].

Theis et al.’s [26] cross-sectional study was conducted to examine the impact of COVD-19 pandemic-related limitations on physical activity and mental health in children and adolescents. The electronic survey was conducted from June to July 2020. Respondents reported negative effects of isolation, with 61% reporting a decline in physical activity levels and over 90% reporting a negative impact on mental health, including poorer behavior, mood, fitness, and regression in social activities and learning [26].

Since most studies on COVID-19 virus infection have focused on transmission, morbidity, and mortality, a study by Poudel was conducted with the aim of assessing the impact of the virus on health-related quality of life (HRQoL) [27]. The results of this study suggest that a greater impact on HRQoL was achieved in acute infections, women, elderly patients, patients with more severe disease, and patients from low-income countries [27].

COVID-19 can lead to many permanent symptoms such as fatigue, shortness of breath, and decreased ability to perform activities of daily living [28,29]. Clearly, recovery or rehabilitation is necessary to return to a normal state after infection. Pulmonary rehabilitation is an evidence-based intervention that addresses many of the symptoms experienced by long-term infected individuals of COVID-19 (defined as symptoms that persist ≥ 3 months after infection), such as shortness of breath, low energy, impaired ability to perform daily tasks, sleep disturbances, and lower self-confidence. At the molecular level, fatigue and fatigue after COVID-19 could be related to persistent inflammation [28,29,30,31]. Pandemic lockdowns have been thought to lead to inactivity and an increase in sedentary behavior, and all necessary measures should be taken to prevent these effects. During isolation, people change their lifestyle habits, increasing the time spent sitting and decreasing the time spent exercising [29,30,31,32,33].

In a study conducted in Spain [34] related to students, it was confirmed that the time spent sitting increased, but surprisingly, the time spent in physical activity also increased, as well as the number of days participants were active. It was found that students who ate a Mediterranean diet exercised more, whereas there was little change in their physical activity behavior. This only goes to show that those who lead a healthy lifestyle also pay attention to their diet and exercise and maintain their habits regardless of the environment. Although the results are positive in terms of physical activity, it should be noted that this population could experience health problems in the future due to the increase in sitting. The most likely reason for the increase in physical activity among young people is that they have realized that their time spent sitting has increased (not going to school, shopping, etc.) and have compensated for this by exercising [34,35]. Another reason could be that students are primarily studying in the health sciences, so they were more inclined to exercise during the pandemic than students in other majors, such as engineering [35].

The limitation of this study is the lack of objective methods of measurement, such as laboratory tests before and after COVID-19, but because infection as well as prolongation of symptoms could not be predicted, this limitation must be justified. Further, the measurement of the costs in case of work loss of these individuals affected by the COVID syndrome could be of importance and interesting to evaluate.

## 5. Conclusions

The presence of the disease of COVID-19 in the previous 6 months in working-age participants significantly affected the duration of aerobic, anaerobic, and high-intensity physical activities, but also the ability to perform certain activities such as walking, which is a basic aerobic activity and a measure of overall health in middle-aged participants. In the majority of cases (78%), participants reported a decrease in educational and productive activities after COVID-19 compared with the period before COVID-19. Statistical analysis of dietary habits, such as consumption of carbohydrates, proteins, vitamins, minerals, and fatty foods, showed that there was no statistically significant difference in the consumption of these nutrients before and after infection. Health policies should be further guided by these findings.

## Figures and Tables

**Figure 1 jcm-11-06668-f001:**
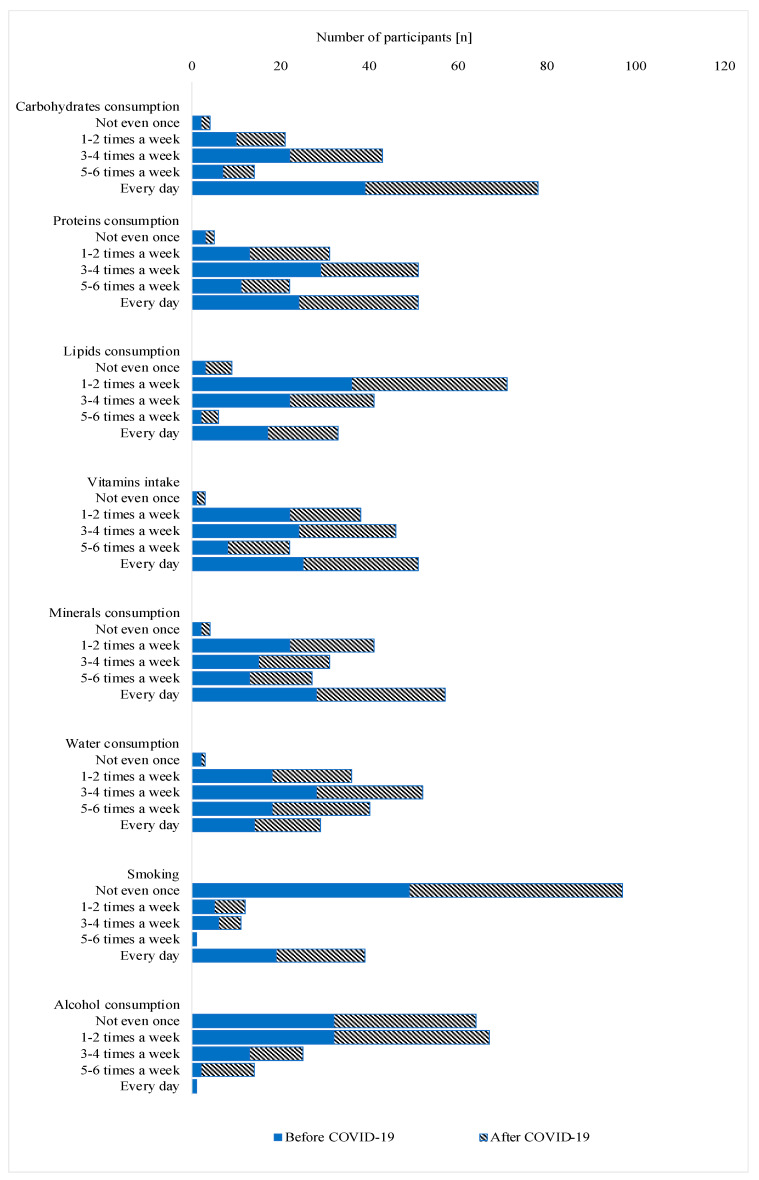
Assessment of weekly food intake and dietary habits of participants before and 6 months after COVID-19. Results are presented as number of patients out of total (*n* = 80).

**Table 1 jcm-11-06668-t001:** Basic socio-economic characteristics of the study population (*n* = 80). Results are presented as distributions [%]/mean ± stand. deviation.

Variables	Categories (*n* = 80)	Distribution [%]/Mean ± SD
Gender [%]	Male	20 [25%]
	Female	60 [75%]
Age category [%]	20–29 years old	54 [68%]
	30–39 years old	9 [12%]
	40–49 years old	17 [20%]
Age [years; X ± SD]	30.64 ± 1.54	
Education level [%]	Elementary school	25 [32.9%]
	High school	6 [2.6%]
	Faculty	21 [27.6%]
	Student	28 [36.8%]
Marital status [%]	Single	33 [43.4%]
	Married	23 [30.3%]
	Bisexual/homosexual relationship	21 [22.4%]
	Divorced	2 [2.6%]
	Widow/er	1 [1.3%]
Place of living [%]	Big city (above 500,000 inhabitants)	10 [7.9%]
	Middle city (100,000–500,000 inhabitants)	70 [82.1%]
Occupation level [%]	Employed	80 [100%]

**Table 2 jcm-11-06668-t002:** Post-COVID-19 clinical status of participants (*n* = 76). Results are presented as frequency of responses [%]. Statistical significance was obtained using the Chi-square test (X^2^). Single or double asterisks represents the statistically significant difference (* *p* < 0.05; ** *p* < 0.01) in distribution of responses among study population.

Health Problems	Severity of Clinical Problem	Frequency [%]	Do the Above Problems Still Exist?		*p* Values
			Yes (%)	No (%)	
Sleep problems (this includes problems falling asleep, sleeping through the night, and waking up early)	None	52 (68.4%)	11 (14.5%)	65 (85.5%)	0.021 *
	Mild problem	8 (10.5%)			
	Moderate problem	11 (14.5%)			
	Big problem	5 (6.6%)			
	Extreme problem	0 (0%)			
Bowel dysfunction (e.g., diarrhea and constipation)	None	56 (73.7%)	4 (5.3%)	72 (94.7%)	0.688
	Mild problem	13 (17.1%)			
	Moderate problem	6 (7.9%)			
	Big problem	1 (1.3)			
	Extreme problem	0 (0%)			
Bladder dysfunction (e.g., incontinence, kidney or bladder stones, kidney problem, urine leakage, and reflux)	None	68 (89.5%)	3 (3.9%)	71 (96.1%)	0.216
	Mild problem	4 (5.3%)			
	Moderate problem	3 (3.9%)			
	Big problem	0 (0%)			
	Extreme problem	1 (1.3%)			
Limitation of movement (restricted range of motion of the joints)	None	45 (59.2%)	6 (7.9%)	70 (92.1%)	0.275
	Mild problem	16 (21.1%)			
	Moderate problem	11 (14.5%)			
	Big problem	2 (2.6%)			
	Extreme problem	2 (2.6%)			
Muscle problems (uncontrolled, spasmodic muscle movements, such as uncontrollable twitching or spasm)	None	48 (63.2%)	8 (10.5%)	68 (89.5%)	0.216
	Mild problem	16 (21.1%)			
	Moderate problem	10 (13.2%)			
	Big problem	1 (1.3%)			
	Extreme problem	1 (1.3%)			
Respiratory problems (difficulty breathing and increased secretion)	None	32 (42.1%)	11 (14.5%)	65 (85.5%)	0.005 **
	Mild problem	21 (27.6%)			
	Moderate problem	13 (17.1%)			
	Big problem	8 (10.5%)			
	Extreme problem	2 (2.6%)			
Limitation of the sense of smell (e.g., reduced or increased perception of smell)	None	33 (43.4%)	13 (17.1%)	63 (82.9%)	0.021 *
	Mild problem	16 (21.1%)			
	Moderate problem	8 (10.5%)			
	Big problem	9 (11.8%)			
	Extreme problem	10 (13.2%)			
Limitations of the sense of taste (e.g., reduced or increased perception of taste)	None	38 (50%)	12 (15.8%)	64 (84.2%)	0.026 *
	Mild problem	14 (18.4%)			
	Moderate problem	8 (10.5%)			
	Big problem	7 (9.2%)			
	Extreme problem	9 (11.8%)			
Circulation problems or circulatory disorders (including swelling of veins, feet, hands, legs)	None	62 (81.6%)	4 (5.3%)	72 (94.7%)	0.125
	Mild problem	10 (13.2%)			
	Moderate problem	3 (3.9%)			
	Big problem	1 (1.3%)			
	Extreme problem	0 (0%)			
Malaise (e.g., increased fatigue)	None	19 (25%)	18 (23.7%)	58 (76.3%)	0.003 **
	Mild problem	16 (21.1%)			
	Moderate problem	16 (21.1%)			
	Big problem	13 (17.1%)			
	Extreme problem	12 (15.8%)			
Fears and anxiety	None	50 (65.8%)	9 (11.8%)	67 (88.2%)	0.043 *
	Mild problem	14 (18.4%)			
	Moderate problem	5 (6.6%)			
	Big problem	5 (6.6%)			
	Extreme problem	2 (2.6%)			
Pain	None	40 (52.6%)	6 (7.9%)	70 (82.1%)	0.162
	Mild problem	18 (23.7%)			
	Moderate problem	10 (13.2%)			
	Big problem	4 (5.3%)			
	Extreme problem	4 (5.3%)			

**Table 3 jcm-11-06668-t003:** Differences in the frequency of aerobic and anaerobic physical activity, as well as in low-, middle-, and high-intensity activity before and six months after COVID-19. Results are presented as frequency in percent from total number of participants [%]. Statistical significance was obtained using Chi-square test (X2) as follows: ^a^ = statistically significantly higher number of participants compared to activity before and after COVID-19, ^b^ = statistically significantly lower number of participants compared to activity before and after COVID-19.

Type of Activity		Aerobic Activity	Anaerobic Activity	Low-Intensity Physical Activity	Middle-Intensity Physical Activity	High-Intensity Physical Activity
Period	Frequency	Percent [%]	Percent [%]	Percent [%]	Percent [%]	Percent [%]
Exercise before COVID-19	Neither day	48.8	56.3	20.0	51.2	50.0
	1–2 days per week	32.5	22.5	23.8	28.7	33.8
	3–4 days per week	11.3	15.0	30.0	15.0	10.0
	5–6 days per week	2.5	5.0	6.3	1.3	5.0
	Every day	5.0	1.3	20.0	3.8	1.3
Exercise after COVID-19	Neither day	60.0 ^a^	62.5 ^a^	16.3	53.8	56.3 ^a^
	1–2 days per week	22.5 ^b^	22.5	36.3 ^a^	32.5	30.0 ^b^
	3–4 days per week	12.5	10.0 ^b^	25.0	75.0 ^a^	6.3
	5–6 days per week	2.5	2.5	8.8	3.8	6.3 ^a^
	Every day	2.5 ^b^	2.5	13.8 ^b^	2.5	1.3
Duration of physical exercise before COVID-19	None	46.3	53.8	12.5	51.2	46.3
	10–20 min	22.5	18.8	16.3	15.0	11.3
	21–40 min	16.3	17.5	25.0	16.3	23.8
	41–60 min	7.5	6.3	18.8	6.3	10.0
	Above 60 min	7.5	3.8	27.5	11.3	8.8
Duration of physical exercise after COVID-19	None	55 ^a^	61.3 ^a^	16.3 ^a^	51.2	57.5 ^a^
	10–20 min	23.8	22.5	20.0	25.0	17.5
	21–40 min	11.3 ^b^	7.5 ^b^	25.0	11.3 ^b^	11.3
	41–60 min	2.5 ^b^	5.0	17.3	6.3	10.0 ^b^
	Above 60 min	7.5	3.8	21.3	6.3	3.8
Possibility of doing exercises before infection	Always less than I wanted	51.2	51.2	25.0	45.0	42.5
	Sometimes less than I wanted	12.5	12.5	6.3	10.0	10.0
	As much as I could	31.3	33.8	46.3	37.5	42.5
	Sometimes more than I wanted	2.5	0	13.8	1.3	1.3
	Always more than I wanted	2.5	2.5	8.8	6.3	3.8
Possibility of doing exercises after infection	Always less than I wanted	57.5 ^a^	52.5	26.3	46.3	52.5 ^a^
	Sometimes less than I wanted	8.8 ^b^	12.5	10.0 ^b^	15.0 ^b^	10.0
	As much as I could	27.5 ^b^	30.0 ^b^	43.8	35.0 ^b^	31.3 ^b^
	Sometimes more than I wanted	1.3	1.3	13.8	1.3	2.5
	Always more than I wanted	5.0	3.8	6.3	2.5	3.8

**Table 4 jcm-11-06668-t004:** Participation in different activities in the period before and six months after COVID-19 among the study population. Statistical significance was obtained using the Chi-square test (X^2^). Asterisks (*) represent the statistically significant difference in selected characteristic of activity before and six months after COVID-19, and the double asterisk (**) represents high significance (<0.001).

Activity and Its Characteristics (*n* = 80)		*p* Values
Daily activity	Frequency [days per week]	0.000 **
	Duration of activity [hours]	0.000 **
	Ability to perform [yes/no]	0.345
Other free-daily activity	Frequency [days per week]	0.013 *
	Duration of activity [hours]	0.040 *
	Ability to perform [yes/no]	0.987
Social activities	Frequency [days per week]	0.016 *
	Duration of activity [hours]	0.001 **
	Ability to perform [yes/no]	0.012 *
Productive activities	Frequency [days per week]	0.081
	Duration of activity [hours]	0.000 **
	Ability to perform [yes/no]	0.670
Educational activities	Frequency [days per week]	0.023 *
	Duration of activity [hours]	0.222
	Ability to perform [yes/no]	0.035 *
Sleeping/Resting	Duration of activity [hours]	0.662
	Ability to perform [yes/no]	1.000

**Table 5 jcm-11-06668-t005:** Change in dietary habits in relation to frequency of consumption per week in the study population. Results were obtained using the Chi-square test. Single asterisks represent the statistically significant difference (* *p* < 0.05) in distribution of responses among study population.

Weekly Intake [Times Per Week]	*p* Values
Carbohydrates	0.827
Proteins	0.564
Lipids	0.251
Vitamins	0.128
Minerals	0.157
Water	0.049 *
Cigarettes	0.987
Alcohol	0.788

**Table 6 jcm-11-06668-t006:** Quality of life of participants six months after COVID-19. Results are presented as frequency of responses [%]. Statistical significance was obtained using the Chi-square test (X2). Asterisks (*) represent the statistically significant difference in distribution of responses among study population (*p* < 0.05).

Item	Graded Responses	Frequency [%]	*p* Values
How would you rate the quality of your life?	Very satisfied	18 (23,7%)	0.045 *
	Satisfied	52 (68,4%)	
	Neither satisfied nor dissatisfied	6 (7,9%)	
	Dissatisfied	0 (0%)	
	Very dissatisfied	0 (0%)	
How satisfied are you with your health?	Very satisfied	15 (19.7%)	0.035 *
	Satisfied	53 (69.7%)	
	Neither satisfied nor dissatisfied	8 (10.5%)	
	Dissatisfied	0 (0%)	
	Very dissatisfied	0 (0%)	
How satisfied are you with your ability to perform daily life activities?	Very satisfied	26 (34.2%)	0.039 *
	Satisfied	42 (55.3%)	
	Neither satisfied nor dissatisfied	8 (10.5%)	
	Dissatisfied	0 (0%)	
	Very dissatisfied	0 (0%)	
How satisfied are you with yourself?	Very satisfied	29 (38.2%)	0.055
	Satisfied	40 (52.6%)	
	Neither satisfied nor dissatisfied	7 (9.2%)	
	Dissatisfied	0 (0%)	
	Very dissatisfied	0 (0%)	
How satisfied are you with your personal relationships?	Very satisfied	24 (31.6%)	0.028 *
	Satisfied	44 (57.9%)	
	Neither satisfied nor dissatisfied	8 (10.5%)	
	Dissatisfied	0 (0%)	
	Very dissatisfied	0 (0%)	
How satisfied are you with the conditions of your housing?	Very satisfied	34 (44.7%)	0.571
	Satisfied	37 (48.7%)	
	Neither satisfied nor dissatisfied	5 (6.6%)	
	Dissatisfied	0 (0%)	
	Very dissatisfied	0 (0%)	

## Data Availability

All data are available on request.

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
