# Peer review of "Post-COVID-19 Status and Its Physical, Nutritional, Psychological, and Social Effects in Working-Age Adults—A Prospective Questionnaire Study"

_jcm, 2022, doi:10.3390/jcm11226668_

Round 1
Reviewer 1 Report
Dear Author, thank you for sending this paper. Minor wrros I have found:
Errors: Most participants were female, not male.
Big city, middle city - what does it mean?
Author Response
R1
Dear Author, thank you for sending this paper. Minor wrros I have found:
Errors: Most participants were female, not male.
A: Thank you for these comments. Corrected.
Big city, middle city - what does it mean?
A: Thank you for these comments. Corrected.
R2
It is an attempt to shed some lights on the functional and cognitive status in actively employed people following Covid-19 disease. The article is promising but it is difficult to read due to lack of fluency. The English language needs significant re-polishing to deliver the message in a more impactful manner.
A: Thank you for these comments. Corrected. Please see a certificate of lecture.
I notice that you included only individuals between 18 and 50 years old. What about fully employed individuals in their sixties?
A: Thank you for these comments. The main exclusion criteria was age above 50, because previous literature data suggested that it is borderline when usually starts many chronical diseases, and reason for the hospital admission during COVID-19. Also, age 50 was chosen as the knot for the regression models (reference: Davis JW, Chung R, Juarez DT. Prevalence of comorbid conditions with aging among patients with diabetes and cardiovascular disease. Hawaii Med J. 2011 Oct;70(10):209-13.) By excluding these factors, we provided healthy and homogenous study group.
A questionnaire may be initially appropriate but are you considering further follow up of these individuals?
A: Thank you for these comments. Unfortunately, we did not plan to follow them further by this way. But it is recommended to all of them that in case of any symptom or new condition or habit to inform their doctors.
Are you completely sure that these are all healthy individuals? It would be more appropriate to establish more explicit criteria for assessment.
A: Thank you for these comments. According to the inclusion criteria, we included only healthy participants (working-age) with no other any chronical disease or treatment in general. We explained that in the section Inclusion/Exclusion criteria.
The discussion is quite disorganised and not really focused on your findings. What are your thoughts? How do your findings fit in the context of other observational studies? What are you proposing?Although I commend your efforts to address such an important matter, I think the article would benefit from a complete rearrangement of its structure.
A: Thank you for these comments. We corrected it completely. Please see section Discussion.
R3
The subject of the work is very important. The knowledge of behavior and lifestyle changes caused by the COVID-19 pandemic is an important issue that allows for more precise and personalized treatment of patients.
A:Thank you for these opinion.
In order to increase the quality and impact of the manuscript, I believe that a few corrections could be made.
- it is a pity that the population is so small. In view of such high morbidity, even assuming age criteria, a larger population means more reliable results. It's just a note, I understand it's hard to change the size of the population right now
A:Thank you for these comments, we are absolutely agree with you. We tried during the pandemic conditions to collect much more homogenous “younger” population with no other present comorbidity and to analyze the many lifestyle aspects in the post covid status. The focus was on working age population, and also since that in our low-economy countries is small rate of employed peoples, that is another reason why the study group “smaller”. Thank you for understanding.
- the age criterion from 18 to 50 years. While the exclusion of children (and young people) is logical, we are talking about working people, I do not understand why the 50-year-old criterion was set. After all, people aged 50 75 also work very intensively, are physically active, take part in marathons, etc. In the chapter on criteria, please include a sentence explaining the reasons for such a limitation.
A: Thank you for these comments. We corrected that. The main exclusion criteria was age above 50, because previous literature data suggested that it is borderline when usually starts many chronical diseases, and reason for the hospital admission during COVID-19. Also, age 50 was chosen as the knot for the regression models (reference: Davis JW, Chung R, Juarez DT. Prevalence of comorbid conditions with aging among patients with diabetes and cardiovascular disease. Hawaii Med J. 2011 Oct;70(10):209-13.) By excluding these factors, we provided healthy and homogenous study group.
- the terms "big city" and "middle city" appear in work and in tables. What does it mean? there were no people from the village? all from the city, but not "small" :-) I propose to replace it by specifying the size of the town, eg above 100,000 inhabitants and below.
A: Thank you for these comments. We corrected that. Please see the Table 1 where is noted that Big city (above 500,000 inhabitants) and Middle city (100,000-500,000 inhabitants), It was important to define the place of living and inhabitants from rural places and from other regions of Serbia (North and South Serbia) because of the different habits in lifestyle and nutrition, as well as in physical activity.
- While describing the marital status in Table 1, typical terms were used, but in the work I did not find information on how homosexuals were classified. Are partners in a stable relationship combined a marriage or two single people? Please include an explanation in the appropriate chapter.
A:Thank you these comments. We corrected it. Homosexual relationship is added as category in the line with bisexual, since that all homosexual people are counted there. Unfortunately in our country homosexual marriage are not approved, we added this information in “relationship category”, please see Table 1.
- I was struck by the phrase "water consumption". Especially the thresholds. According to Fig 1 some patients drink water once a week? so like a camel? Of course, the idea was to consume water extra. it is very difficult to distinguish between drinking coffee, teas and herbal infusions, soups and similar liquids from additional water consumption. I think the methodology used allowed for the reliable consumption of this "extra" water before falling ill, but it is necessary to describe the criterion in detail so that there is no doubt.
A: Thank you for these comments. In general, the National Academies of Sciences, Engineering, and Medicine suggest that each day women get a total of about 2.7 liters (L), or 11 cups, of fluid and men get about 3.7 L (16 cups). Every additional cup is recognized as additional water consumption.
R4
Dear authors, the document is very well structured, the methodology is appropriate and the results are correctly described and adequately discussed.
A: Thank you very much for these nice comments. We really appreciate it.
In line 386, delete 6. Patents
A: Thank you for these comments. We corrected that.

Reviewer 2 Report
It is an attempt to shed some lights on the functional and cognitive status in actively employed people following Covid-19 disease. The article is promising but it is difficult to read due to lack of fluency. The English language needs significant re-polishing to deliver the message in a more impactful manner.
I notice that you included only individuals between 18 and 50 years old. What about fully employed individuals in their sixties? A questionnaire may be initially appropriate but are you considering further follow up of these individuals? Are you completely sure that these are all healthy individuals? It would be more appropriate to establish more explicit criteria for assessment. The discussion is quite disorganised and not really focused on your findings. What are your thoughts? How do your findings fit in the context of other observational studies? What are you proposing?
Although I commend your efforts to address such an important matter, I think the article would benefit from a complete rearrangement of its structure.
Author Response

(The authors gave the same response as above.)

Reviewer 3 Report
The subject of the work is very important. The knowledge of behavior and lifestyle changes caused by the COVID-19 pandemic is an important issue that allows for more precise and personalized treatment of patients.
In order to increase the quality and impact of the manuscript, I believe that a few corrections could be made.
- it is a pity that the population is so small. In view of such high morbidity, even assuming age criteria, a larger population means more reliable results. It's just a note, I understand it's hard to change the size of the population right now
- the age criterion from 18 to 50 years. While the exclusion of children (and young people) is logical, we are talking about working people, I do not understand why the 50-year-old criterion was set. After all, people aged 50 75 also work very intensively, are physically active, take part in marathons, etc. In the chapter on criteria, please include a sentence explaining the reasons for such a limitation.
- the terms "big city" and "middle city" appear in work and in tables. What does it mean? there were no people from the village? all from the city, but not "small" :-) I propose to replace it by specifying the size of the town, eg above 100,000 inhabitants and below.
- While describing the marital status in Table 1, typical terms were used, but in the work I did not find information on how homosexuals were classified. Are partners in a stable relationship combined a marriage or two single people? Please include an explanation in the appropriate chapter.
- I was struck by the phrase "water consumption". Especially the thresholds. According to Fig 1 some patients drink water once a week? so like a camel? Of course, the idea was to consume water extra. it is very difficult to distinguish between drinking coffee, teas and herbal infusions, soups and similar liquids from additional water consumption. I think the methodology used allowed for the reliable consumption of this "extra" water before falling ill, but it is necessary to describe the criterion in detail so that there is no doubt.
Author Response

(The authors gave the same response as above.)

Round 2
Reviewer 2 Report
No further comments.
Reviewer 3 Report
I believe the manuscript is now ready for publication. I have no additional comments